# Prevalence and risk factors of hypertension among adults: A community based study in Addis Ababa, Ethiopia

**Meseret Molla Asemu**[1]*, **Alemayehu Worku Yalew**[1], **Negussie Deyessa Kabeta**[1], **Desalew Mekonnen**[2]

**1** School of Public Health, Addis Ababa University, Addis Ababa, Ethiopia, **2** College of Health Science, Addis Ababa University, Addis Ababa, Ethiopia

* mesinebi@yahoo.com, meseretmollakr@gmail.com

**Data Availability Statement:** All relevant data are within the manuscript and its Supporting Information files.

## Abstract

### Background

In all areas of the World Health Organization, the prevalence of hypertension was highest in Africa. High blood pressure is a significant risk factor for coronary and ischemic diseases, as well as hemorrhagic stroke. However, there were scarce data concerning the magnitude and risk factors of hypertension. Thus, this study aimed to identify the prevalence and associated factors of hypertension among adults in Addis Ababa city.

### Method

A community-based cross-sectional study was conducted from June to October 2018 in Addis Ababa city. Participants aged 18 years and older recruited using a multi-stage random sampling technique. Data were collected by face-to-face interview technique. All three WHO STEPS instruments were applied. Additionally, participants' weight, height, waist, hip, and blood pressure (BP) were measured according to standard procedures.

Multiple logistic regressions were used and Odds ratios with 95% confidence intervals were also calculated to identify associated factors.

### Results

In this study, a total of 3560 participants were included. The median age was 32 years (IQR 25, 45). More than half (57.3%) of the respondents were females. Almost all (96.2%) of participants consumed vegetables and or fruits less than five times per day. Eight hundred and sixty-five (24.3%) of respondents were overweight, while 287 (8.1%) were obese. One thousand forty-one 29.24% (95% CI: 27.75–30.74) were hypertensive, of whom two-thirds (61.95%) did not know that they had hypertension.

Factors significantly associated with hypertension were age 30–49 and ≥50 years (AOR = 2.79, 95% CI: 1.39–5.56) and (AOR = 8.23, 95% CI: 4.09–16.55) respectively, being male (AOR = 1.88, 95% CI: 1.18–2.99), consumed vegetables less than or equal to 3 days per week (AOR = 2.44, 95% CI: 1.21–4.93), obesity (AOR = 2.05, 95%CI: 1.13–3.71),

**Funding:** This work was funded by Addis Ababa University, Addis Ababa, Ethiopia. The funders had no role in study design, data collection and analysis, decision to publish, or preparation of the manuscript.

**Competing interests:** The authors have declared that no competing interests exist.

abdominal obesity (AOR = 1.70, 95% CI: 1.10–2.64) and high triglyceride level (AOR = 2.06, 95% CI: 1.38–3.07).

## Conclusion

In Addis Ababa, around one in three adults are hypertensive. With a large proportion, unaware of their condition. We recommend integrating regular community-based screening programs as integral parts of the health promotion and disease prevention strategies. Life-style interventions shall target the modifiable risk factors associated with hypertension, such as weight loss and increased vegetable consumption.

## Introduction

Between 1980 and 2010, the proportion of the world's population with high blood pressure (defined as systolic and or diastolic blood pressure $\geq$ 140/90 mmHg) or uncontrolled hypertension had dropped modestly. However, sharp rises due to population growth and aging have been recorded across the World Health Organization (WHO) regions over the past decade, with the largest rise in Africa at 30%. The lowest prevalence of raised blood pressure was noted in the Americas region, at 18%, while the global estimate among adults aged 18 years and above was around 22% in 2014. According to the WHO estimates,Ethiopia tops at 24.4% for all adults combined [1–3].

High blood pressure accounts for about 13.5% of annual deaths in the world. Moreover, hypertension directly accounts for 54% of all strokes and 47% of all coronary artery disease worldwide. At the same time, the most productive segment of the population is those aged 45 to 69, who make up more than half of this burden [4].

High blood pressure is a major risk factor for coronary and ischaemic diseases as well as bleeding stroke. It has been shown that blood pressure levels are positively associated with the risk of stroke and coronary heart disease [5]. One of the most modifiable risk factors for cardiovascular diseases is hypertension. However, awareness towards the treatment and control of hypertension is extremely low among the low and middle-income counties (LMICs), including Ethiopia.On top of this, the health care resources of the LMICs are overwhelmed by other priorities, including HIV/AIDS, tuberculosis, and malaria. As a result, many LMICs have not yet given due attention to its prevention and control [6].

In Ethiopia, non-communicable diseases such as hypertension and diabetes mellitus have begun to emerge as the leading causes of hospital admissions, morbidity, and mortality in health facilities located around the nation [7]. A 2016 report by the Ethiopian Public Health Institute (EPHI) found that 95% of Ethiopian adult populations have 1 to 2 risk factors for non-communicable diseases [8,9]. But there was little information on the extent and risk factors for hypertension at the community level in Ethiopia, including the Addis Ababa study area.

That little information was done by using the WHO stepwise tool step one and step two only [6,10]. And the study setting was at the facility level, though; there was a single study done at the national level using all the three World Health Organization stepwise tools [8,11]. Besides, the study area, Addis Ababa, is the largest urban center and capital of Ethiopia, providing approximately one-quarter of the urban population in Ethiopia [6]. This study aimed to determine the prevalence and associated factors of hypertension in the adult population of Addis Ababa using the three stepwise tools of the World Health Organization.

## Methods

### Study design and area

A cross-sectional community study was conducted from 1 June to 31 October 2018 in Addis Ababa City. Addis Ababa city is the capital city of Ethiopia. Administratively, Addis Ababa subdivided into ten sub-cities and 116 woredas [12]. According to the Central Statistical Agency of the Federal Democratic Republic of Ethiopia, the city was projected to inhabit 3,433,999 population by 2017 [13].

### Sampling techniques and sample size determination

Multi-stage cluster sampling techniques were employed by first identifying seven of the ten sub-cities based on preset criteria, including the location of the area, population density, and socioeconomic status. Then, one woreda was randomly selected from each selected sub-cities. After that, two 'ketenas' were randomly picked from the chosen woredas, which are the smallest geographical units within woredas. Finally, for each ketena, the first household was randomly selected, while subsequent households were selected based on proximity to the first and the preceding household.

A total of 3,724 eligible adults aged 18 and over were interviewed at the selected households. The required sample size was determined using the single population proportion formula by considering: prevalence of hypertension 31.5% from a previous study done in Addis Ababa, Ethiopia [6], α = 0.05 (z = 1.96), the margin of error 2%, design effect of 1.5 and 20% possible non-response rate. We also determined the sample size for the risk factors of hypertension by using two population proportion formula. But the maximum sample size was attained during the single population proportion formula. As well, the total sample size for each sub-city was determined using with probability proportional to size (PPS).

### Data collection instruments and measurements

We used the adapted WHO STEPwise approach to surveillance tools. These tools have a sequential process and aim to serve as an entry point for low- and middle-income countries to monitor chronic diseases and their risk factors. All the three WHO STEPS instrument was applied to collect data on the selected information, including socio-demographic, behavioral, physical, and biochemical measurements as a part of the core and expanded modules [14]. The tools were first pretested among adults found outside the study area and, then modifications were made based on the findings.

The data were collected via face-to-face interview by trained baccalaureate nurse and laboratory technicians. Weighing scales and non stretch tape were used to measure body weight and height. Weight and height were measured as participants were standing without shoes and wearing lightweight clothing. Height was recorded to the nearest 0.5 cm; weight was recorded to the nearest 100g. Body Mass Index (BMI) was calculated as weight in kilograms divided by height in meters squared (weight (kg)/height (m$^2$) and classified as underweight (<18.5), normal (18.5–24.9), overweight (25–29.9) and obese ($\geq$ 30.0).

Waist circumference was measured at the level of the iliac crest using a non stretch tape measure. Hip circumference measured at the maximum circumference of the hip and; waist-to-hip ratio (WHR) calculated as a ratio of waist and hip circumference.

Physical activity was measured using the Global Physical Activity Questionnaire (GPAQ) section of the STEPS instrument, and the total physical activity is presented in MET (metabolic equivalent) minutes per week. The instrument explores three main areas of day-to-day activities: work (including domestic work), transport, and recreational activities. The level of total

physical activity was subsequently classified into high, moderate, or low using the GPAQ analysis guideline provided along with the STEPS instrument [14].

Using a standardized automated blood pressure monitor, blood pressure was measured on the left arm as per the WHO protocol by informing the participants to remain seated and relaxed.Three blood pressure measurements were taken with at least 3-minute intervals between them. The mean value of the 2nd and 3rd measurements was used for analysis [14]. Blood pressure (BP) classified according to the Seventh Joint National Committee on Prevention, Detection, Evaluation, and Treatment of High Blood Pressure (JNC VII) [3].

To ensure the quality of the data collection, data collectors were trained by the principal investigator; and later on, daily checks were carried out by field supervisors and the principal investigator. The weight of the participants, measured on a pre-calibrated electronic scale. Weighing scales checked and zero levels adjusted between measurements; we also placed the scale on a firm flat surface. The blood pressure was measured in a seated position by a digital device (OMRON M2 Eco). The instrument has been clinically approved and recommended by the World Health Organization. In addition, WHO's STEPwise tools have been previously validated and implemented in mainly developing countries, including Ethiopia [6,8].

## Operational definitions

**Hypertension**: defined as a mean measured blood pressure of $\geq$ 140 mmHg systolic and/or the mean measured diastolic blood pressure of $\geq$ 90 mmHg or self-reported history of hypertension.

**Body Mass Index (BMI)**: calculated as weight in kilograms divided by height in meters squared (weight (kg)/height (m$^2$). BMI was categorized as per the World Health Organization guidelines [14], underweight (BMI <18.5), normal (BMI $\geq$18.5 to $\leq$ 24.9), overweight (BMI $\geq$ 25.0 to $\leq$ 29.9) or obese (BMI $\geq$ 30.0).

**Waist to hip ratio**: calculated as waist circumference in cm divided by hip circumference in cm and it was used as a measure of abdominal obesity. Waist to hip ratio $\geq$ 0.90 m in men and $\geq$ 0.85m in women is defined as having abdominal obesity [15].

**High physical activity:** a person reaching any of the following criteria is classified in this category:

- Vigorous-intensity activity on at least 3 days achieving a minimum of at least 1,500 MET-minutes/week OR

- 7 or more days of any combination of walking, moderate- or vigorous intensity activities achieving a minimum of at least 3,000 MET-minutes per week.

**Moderate physical activity:** a person not meeting the criteria for the "high" category, but meeting any of the following criteria is classified in this category:

- 3 or more days of vigorous-intensity activity of at least 20 minutes per day

  OR

- 5 or more days of moderate-intensity activity or walking of at least 30 minutes per day OR

- 5 or more days of any combination of walking, moderate- or vigorous intensity activities achieving a minimum of at least 600 MET-minutes per week.

**Low physical activity:** a person not meeting any of the above mentioned criteria under moderate or high physical activities falls in this category.

**Raised fasting blood glucose** was defined as capillary whole blood value $\geq$110 mg/dl.

**Raised total cholesterol** was defined as total blood cholesterol level $\geq$190mg/dl.

**Raised triglyceride was** defined as raised triglyceride level ≥150 mg/dl.

## Data analysis

Double data entry procedures were performed using the EpiData 3.1 statistical software, and analyses were performed using IBM SPSS software version23. Binary logistical regression was used to identify risk factors for hypertension. Initially, possible risk factors were assessed using bivariate analyses; then we did the multivariable logistic regression model to control confounding factors, and statistical significance was accepted when the P-value < 0.05. The Hosmer-Lemeshow goodness-of-fit statistic was used to evaluate whether or not the assumptions necessary for the application of multiple logistic regression are met. Odds ratios (OR) with 95% Confidence Intervals (CI) were computed.

## Ethical clearance

Ethical clearance was obtained from the Addis Ababa University, College of Health Sciences Institutional Review Board (IRB), and the city government of Addis Ababa Health Bureau Ethical Review Committee (ERC). A letter of permission was obtained from the selected sub-city health offices. Respondents were fully informed about the purpose of the study and gave verbal and written consent. Participants having high blood pressure, high blood glucose level, and or abnormal lipid profiles during the study period were referred and informed to go to nearby health facilities for further diagnosis and management.

## Results and discussion

### Description of the study participants

From the total 3724 sampled population, consent was given to the 3560 participants to involve in step one and two questionnaires, making an overall response rate of 95.59%. Using a random sampling technique, 582 (20%) of the study participants who participated in the interview and physical measurements were selected for the step three questionnaires (biochemical assessment).

Respondents were between 18 and 95 years old and, the median age was 32 years old (IQR 25, 45). More than half (57.3%) of the respondents were females. The majority (74.8%) were Orthodox Christians, followed by Muslims (14.9%). Above one-third (37%) of them were self-employed, while nearly a half (49.6%) were currently married (Table 1).

### Behavioral risk factors of the study participants

**Tobacco use.**  Tobacco use was assessed by interviewing respondents about their current smoking status, previous smoking experience, the age they started smoking, and exposure to second-hand smoke. Overall, about 4.2% (150) of survey respondents were current smokers (daily smokers and non-daily smokers) (Table 2). Of these, a majority (88.66%) smoke cigarettes on daily basis, with an average of 10 cigarettes per day. More than three-fourth 136 (90.66%) of current smokers were male compared to female (p < 0.001). The average age at which smokers started smoking was 21 ± 6.58 years. Fifty-five (1.61 percent) have smoked cigarettes in the past. One hundred nineteen (3.4%) were passive smokers or second-hand smokers.

**Khat chewing.**  There were 330 (9.3%) participants who reported chewing khat (Table 2). Over a third of the respondents, 105 (31.53%) and half, 172 (51.65%) chew khat on a daily and weekly basis, respectively. On the other hand, 153 participants (4.3%) had previously chewed khat.

**Table 1. Socio-demographic characteristics of the study participants in Addis Ababa, Ethiopia, October 2018.**

| Characteristics | Frequency | Percent |
|---|---|---|
| **Sex** | | |
| Male | 1520 | 42.7 |
| Female | 2040 | 57.3 |
| **Religion** | | |
| Orthodox | 2664 | 74.8 |
| Muslim | 530 | 14.9 |
| Protestant | 333 | 9.4 |
| Catholic | 14 | 0.4 |
| Other | 19 | 0.6 |
| **Employment status** | | |
| Government employee | 388 | 10.9 |
| Non-government employee | 257 | 7.2 |
| Self employed | 1316 | 37.0 |
| Student | 301 | 8.5 |
| House wife | 750 | 21.1 |
| Daily laborer | 83 | 2.3 |
| Merchant | 69 | 1.9 |
| Unemployed(able to work) | 173 | 4.9 |
| Unemployed(unable to work) | 47 | 1.3 |
| Retired (pensioner) | 176 | 5.0 |
| **Age** | | |
| 18–29 | 1508 | 42.4 |
| 30–49 | 1304 | 36.6 |
| 50 and above (50–95) | 748 | 21.0 |
| **Family size** | | |
| 1–4 | 2178 | 61.2 |
| ≥5 | 1382 | 38.8 |
| **Marital status** | | |
| Never married | 1331 | 37.4 |
| Currently married | 1767 | 49.6 |
| Separated | 49 | 1.4 |
| Divorced | 139 | 3.9 |
| Widowed | 271 | 7.6 |
| Non response | 3 | 0.1 |
| **Highest education level** | | |
| Primary | 1176 | 33.0 |
| Secondary | 719 | 20.2 |
| Preparatory | 464 | 13.0 |
| Technique | 67 | 1.9 |
| College and above | 539 | 15.1 |
| Not attended formal education | 595 | 16.7 |

**Alcohol consumption.** One thousand one hundred sixty-two (32.6%) participants consumed alcohol and of that 783 (22.0%) consumed alcohol during the last month (Table 2). Binge drinking is defined; as consuming alcohol for men five and above, or women four and above drink on one occasion and, the result showed that 269 (7.6%) of men and 81 (2.3%) of women binge drunker.

**Table 2. Prevalence of hypertension across different characteristics of respondents in Addis Ababa city, October 2018.**

| Characteristics | Number | Percent | Hypertension (%, CI) |
|---|---|---|---|
| **Age** | | | |
| 18–29 | 1508 | 42.4 | 12.86 (11.17–14.56) |
| 30–49 | 1304 | 36.6 | 31.29 (28.77–33.81) |
| ≥50 | 748 | 21.0 | 58.69 (55.15–62.23) |
| **Current smoker** | | | |
| Yes | 150 | 4.2 | 34.67 (26.96–42.37) |
| No | 3410 | 95.8 | 29.00 (27.48–30.53) |
| **Current khat use** | | | |
| Yes | 330 | 9.3 | 31.2 (26.19–36.24) |
| No | 3229 | 90.7 | 29.0 (27.45–30.58) |
| **Current alcohol use** | | | |
| Yes | 1162 | 32.6 | 32.79 (30.09–35.49) |
| No | 2397 | 67.4 | 27.53 (25.74–29.32) |
| **Fruit servings consumed in days per week** | | | |
| None | 1107 | 31.1 | 34.24 (31.44–37.04) |
| 1–3 days | 2126 | 59.7 | 27.19 (25.29–29.08) |
| 4–7 days | 266 | 7.5 | 24.81 (19.59–30.04) |
| Don't know | 61 | 1.7 | 29.51 (17.73–41.29) |
| **Vegetable servings consumed in days per week** | | | |
| None | 530 | 14.9 | 30.19 (26.27–34.11) |
| 1–3 days | 2549 | 71.6 | 29.23 (27.46–30.99) |
| 4–7 days | 448 | 12.6 | 28.57 (24.37–32.77) |
| Don't know | 33 | 0.9 | 24.24 (8.81–39.67) |
| **High physical activity** | | | |
| Yes | 542 | 15.2 | 21.77 (18.29–25.26) |
| No | 3018 | 84.8 | 30.58 (28.94–32.23) |
| **Low physical activity** | | | |
| Yes | 986 | 27.7 | 30.53 (27.65–33.41) |
| No | 2574 | 72.3 | 28.75 (27.00–30.50) |
| **Raised blood glucose (≥110 mg/dl)** | | | |
| Yes | 51 | 8.5 | 58.82 (44.84–72.80) |
| No | 551 | 91.5 | 37.57 (33.51–41.62) |
| **Raised cholesterol (≥190 mg/dl)** | | | |
| Yes | 255 | 43.8 | 49.80 (43.63–55.98) |
| No | 327 | 56.2 | 31.50 (26.44–36.56) |
| **Raised triglyceride (≥150 mg/dl)** | | | |
| Yes | 241 | 41.4 | 54.36 (48.02–60.69) |
| No | 341 | 58.6 | 29.03 (24.19–33.87) |
| **Family history of hypertension** | | | |
| Yes | 718 | 20.2 | 38.30 (34.74–41.87) |
| No | 2714 | 76.2 | 26.46 (24.79–28.12) |
| Don't know | 128 | 3.6 | 37.50 (29.00–46.00) |

**Dietary habits of the study participants.** Two thousand three hundred ninety-two (67.2%) consumed fruits, at least one time per week, and the mean fruit consumption per week was 2.12 (±1.48) days, as well, the majority 2268 (94.77%) ate fruit 1–2 serving per day with a mean of 1.36 (±0.58) times per day. Similarly, more than three forth 2997 (84.2%) of the

participants ate vegetables at least one time per week with a mean of 2.46 (± 1.46) per week, and most of them 2827 (94.33%) ate it 1–2 times per day with mean serving time per day was 1.55 (±0.61). Nearly three fourth 2511 (70.6%) of respondents reported that they usually use vegetable oils like Nug (Guizotia Abyssinica), Sesame, and Sunflower oil for meal preparation, while nearly one-fourth 839 (23.6%) use a vegetable oil which was solid at room temperature. Nearly all 3423 participants (96.2%) consumed vegetables and fruits less than five times a day.

**Physical activity.**   The total median physical activity of the respondents was 7440 (IQR 2888, 12240) and, the median total physical activity (TPA, in MET-minutes per week) was estimated to be 7800 (IQR 3000, 14280) in males and 7200 (2880, 11320) in females. Approximately 29.5% of males and 4.6% of females were; categorized as having a high (vigorous) level of TPA. However, significantly more women (36.4%) than men (16.0%) classified as having low levels of TPA (P < 0.001). In addition, most of the study participants, 3198 (89.8%), walked or cycled for a minimum of 10 minutes per day.

## Physiological characteristics of the study participants

**Body mass index and waist to hip ratio.**   Weight and height measured in all participants at 3560; the average BMI for respondents was 23.54 (±4.39 kg/m$^2$). Eight hundred and sixty-five (24.3%) were overweight, while 287 (8.1%) were obese. Moreover, the average hip-waist ratio was 0.88 (±0.086 m$^2$)) with 0.89 (± 0.086 m$^2$) and 0.87 (±0.085 m$^2$) for men and women, respectively. Over two-thirds of females (67.06%) and 32.9% of males had abdominal obesity.

**Biochemical measurements of the respondents.**   Of the total number of participants, the blood sample was collected from 582 participants (20 percent). The average FBS was 86.7 (±36.2 mg/dl), and the prevalence of high blood sugar, high cholesterol, and triglycerides was 8.5%, 43.8%, and 41.4%, respectively (Table 2).

## Prevalence of hypertension

Three consecutive blood pressure measurements took from 3,560 respondents (95.16%) and; an average of the second and third measurements used for blood pressure analysis. The mean systolic and diastolic blood pressure of the respondents was 125.03 (95% CI: 124.39–125.62) mm Hg and 79.58 (95% CI: 79.8–79.97) mm Hg, respectively. The mean SBP was 126.95 (95% CI: 126.03–127.87) mmHg among males and 123.59 (95% CI: 122.72–124.47) mmHg among females. Likewise, the mean DBP was 80.76 (95% CI: 80.14–81.38) mm Hg in males and 78.69 (95% CI: 78.17–79.21) mm Hg in females. Both mean SBP (P < 0.001) and DBP (P < 0.001) were significantly higher in men compared to women.

The overall prevalence of hypertension was 29.24% (95% CI: 27.75–30.74), slightly higher among men 30.13 (95% CI: 27.82–32.44) than women 28.58 (95% CI: 26.66–30.54). Of the 1041 hypertensive respondents, 645 (61.95%) had just been diagnosed in the survey (new screening).

## Factors associated with hypertension

Multivariable logistic regression analysis found that of several non-modifiable factors, age and gender were associated with hypertension. The odds of hypertension increased with increased age. The odds of hypertension increased almost three times AOR = 2.79 (95% CI: 1.39–5.56) among respondents aged 30–49 years, and it was eight times AOR = 8.23 (95% CI: 4.09–16.55) higher among respondents aged 50 years and above as compared to those 18–22 years old. The odds of hypertension were almost twice as high AOR = 1.88 (95% CI: 1.18–2.99) in men compared with women.

From modifiable and other factors, eating fewer vegetables per week, body mass index, abdominal obesity, and high triglycerides levels were associated with hypertension. The odds of hypertension increased more than two times AOR = 2.44 (95% CI: 1.21–4.93) among respondents who consumed vegetable less than or equal to three days per week compared to those who ate more than three days per week.

The chance of hypertension reduced by 73% among underweight participants AOR = 0.27 (95% CI: 0.07–0.97), but the odds were two times higher AOR = 2.05 (95%CI: 1.13–3.71) among obese participants as compared to those having normal BMI. Moreover, the odds of hypertension was almost two times higher AOR = 1.70 (95% CI: 1.10–2.64) among participants with abdominal obesity as compared to their counterparts.

The odds of hypertension was also increased by two AOR = 2.06 (95% CI: 1.38–3.07) among participants who had high triglyceride level as compared to their counterparts.

The odds of hypertension was also increased by two AOR = 2.06 (95% CI: 1.38–3.07) in participants with high triglyceride level compared to their counterparts. In this particular study, risky behaviors, including alcohol use, vigorous physical activity, family history of hypertension or diabetes, high blood sugar, and high cholesterol level not significantly associated with hypertension (Table 3).

## Discussion

The study found that approximately one in three adults aged 18 and over is hypertensive. During childhood, there are modest facts about a gender change in blood pressure. However, beginning with youth, males tend to have a higher average level. But later in life, the difference gets smaller, and the pattern can even be changed [16]. The prevalence of hypertension in the current study is slightly higher among men than women, which is comparable; a community-based study conducted in Addis Ababa, Ethiopia, reported prevalence of hypertension was 31.5% and 28.9% among males and females, respectively [6]. Moreover, this study is also comparable with other community-based studies conducted in Jalalabad, Afghanistan (28.4), Kenya (29.4%), Uganda (30.5%), and Gondar city (28.3%) [10,17–19].

The prevalence of hypertension in this study is considerably higher as compared to other studies Bangladesh (16.0%), Eritrea (16.4%), Addis Ababa (25%), Bahir Dar (25.1%), Durame Southern Ethiopia (22.4%), Gilgel Gibe South West Ethiopia (5.8%) and Mekelle (20.1%) [11,20–25]. The difference may be explained by the age differences of the surveyed populations (18 years and above in our case, whereas in the other studies, the age of the participants varies between 15 and 64 years). Differences may also be attributed to the diversity of sociodemographic characteristics, sample size, lifestyle, and dietary patterns of the study participants.

On the contrary, the prevalence of hypertension in our study is lower than other similar community-based studies conducted in South Africa (38.9%), Sudan (35.7%), Nigeria (33.1%) and Cameroon (47.5%) [26–29]. This disparity can be due to variations in race, genetics, and prevalence of obesity (higher among others), all of which are likely to influence blood pressure.

From non-modifiable risk factors, age is one of the risk factors of hypertension proved by many studies; there is a positive association between age and hypertension when age increases, the odds of hypertension also increases [6,10,11,17,18,21,24,26]. It is primarily due to the increase in systolic blood pressure with age, mainly due to the reduction in elasticity (increased stiffness) of large duct arteries [30]. Inthe same vein, this study, this study found out that respondents aged 30–49 years had 3 times higher odds of hypertension, and 8 fold higher odds among participants aged 50 and above. In terms of gender, the prevalence of hypertension was almost two times higher in males compared to females in the current study, which is consistent with other study findings [11,22,25,28].

**Table 3. Bivariate and multivariable logistic regression analysis of factors associated with hypertension among study participants in Addis Ababa city, October 2018.**

| Variable | Hypertension | | Crude OR (95% CI) | Adjusted OR (95% CI) | P-value |
|---|---|---|---|---|---|
| | Yes | No | | | |
| **Age** | | | | | |
| 18–29 | 194 | 1314 | 1.00 | 1.00 | |
| 30–49 | 408 | 896 | 3.08 (2.55–3.73) | 2.79 (1.39–5.56)* | 0.003 |
| ≥50 | 439 | 309 | 9.62(7.80–11.87) | 8.23 (4.09–16.55)** | < 0.001 |
| **Sex** | | | | | |
| Female | 538 | 1457 | 1.00 | 1.00 | |
| Male | 458 | 1062 | 1.08 (0.93–1.25) | 1.88 (1.18–2.99)* | 0.004 |
| **Education** | | | | | |
| Primary | 334 | 842 | 1.00 | 1.00 | |
| 2$^{ry}$ & preparatory | 304 | 879 | 0.87 (0.73–1.05) | 1.03 (0.58–1.80) | 0.66 |
| Technique & college | 142 | 464 | 0.77 (0.62–0.97) | 0.97 (0.49–1.89) | 0.94 |
| Unable to read & write | 261 | 334 | 1.97 (1.60–2.42) | 1.10 (0.64–1.88) | 0.65 |
| **Alcohol** | | | | | |
| No | 660 | 1737 | 1.00 | 1.00 | |
| Yes | 381 | 781 | 1.28 (1.10–1.49) | 1.35 (0.87–2.09) | 0.34 |
| **High physical activity** | | | | | |
| No | 923 | 2095 | 1.00 | 1.00 | |
| Yes | 118 | 424 | 0.62 (0.51–0.79) | 1.05 (0.51–2.15) | 0.88 |
| **Family history of diabetes** | | | | | |
| Yes | 848 | 2142 | 1.00 | 1.00 | |
| No | 169 | 300 | 0.70 (0.57–0.86) | 1.02 (0.59–1.78) | 0.93 |
| **Family history of hypertension** | | | | | |
| Yes | 718 | 1996 | 1.00 | 1.00 | |
| No | 275 | 443 | 0.58 (0.48–0.69) | 0.73 (0.45–1.19) | 0.34 |
| **Body Mass Index** | | | | | |
| 18.5–24.9 | 504 | 1539 | 1.00 | 1.00 | |
| <18.5 | 48 | 319 | 0.46 (0.33–0.63) | 0.27 (0.07–0.97)* | 0.036 |
| 25–29.9 | 341 | 519 | 2.01 (1.69–2.34) | 1.48 (0.95–2.32) | 0.072 |
| ≥ 30 | 145 | 139 | 3.19 (2.47–4.10) | 2.05 (1.13–3.71)* | 0.011 |
| **Abdominal obesity ≥ 0.90 m (Men) & ≥ 0.85 m (Women)** | | | | | |
| No | 350 | 1308 | 1.00 | 1.00 | |
| Yes | 691 | 1211 | 2.13 (1.84–2.48) | 1.70 (1.10–2.64)* | 0.026 |
| **Raised blood glucose** | | | | | |
| No (< 110 mg/dL) | 207 | 344 | 1.00 | 1.00 | |
| Yes (≥ 110 mg/dL) | 30 | 21 | 2.37 (1.32–4.27) | 0.943 (0.35–2.54) | 0.74 |
| **Raised cholesterol** | | | | | |
| No (<190 mg/dL) | 103 | 224 | 1.00 | 1.00 | |
| Yes (≥ 190 mg/dL) | 127 | 128 | 2.158 (1.54–3.03) | 0.92 (0.46–1.86) | 0.48 |
| **Raised triglyceride** | | | | | |
| No (<150 mg/dL) | 99 | 242 | 1.00 | 1.00 | |
| Yes (≥ 150 mg/dL) | 131 | 110 | 2.91 (2.06–4.11) | 2.06 (1.38–3.07)** | < 0.001 |
| **Vegetable servings consumed in days per week** | | | | | |
| >3 days | 128 | 320 | 1.00 | 1.00 | |
| ≤ 3 days | 905 | 2174 | 1.04 (0.84–1.29) | 2.44 (1.21–4.93)* | 0.009 |

P-value < 0.05 * and <0.000**, (backward logistic regression method was employed).

According to the World Health Organization, overweight and obesity are a major risk factor for heart disease, including high blood pressure, which is the number one cause of death [31]. In our study, the odds of hypertension were two times higher among obese participants compared to those with normal body mass index; however, the chances of hypertension were reduced by 73% among underweight participants. This finding (especially the obese category) was in line with previous reports from Ethiopia, Kenya, Uganda, Sudan, Bangladesh, and Cameroon [10,18,21,22,24,25,27,28,32], and this showed that obesity is one of the risk factor associated with hypertension almost all studies. Moreover, the odds of hypertension were two times higher among abdominally obese respondents compared to their counterparts, the result is also consistent with other studies [27,28,33,34].

Hypertriglyceridemia is a powerful predictor of cardiovascular disease, which causes endothelial damage, and loss of physiological vasomotor activity that results from endothelial damage can occur in the form of high blood pressure [35]. In our study, having a high triglyceride level was independently associated with hypertension. The odds of hypertension increased by two among participants with high triglyceride levels relative to their counterparts; our findings are consistent with those of others [33,36].

Previous studies done so far suggested that use of alcohol, cigarette smoking, Khat use, literacy level, physical activity, raised fasting glucose level, family history of hypertension, family history of diabetes, and excessive salt use were significantly associated with hypertension. In contradiction, in this study, the above variables were not significantly associated with hypertension [4,17,18,24,25,27,37]. A contradicting finding was noted in this current study where all the above variables showed no significant association with hypertension.The inconsistency in these results may be due to the variation of sample size, study settings, and population characteristics. These variations may also be explained by the research design issues (as a cross-sectional design can't distinguish the sequences of explanatory variables and the outcome). The other element of this study included adults 18 years of age or older, but different studies used a different age class, which should make comparisons difficult. Additionally, the respondents might not know whether they had a family history of hypertension or not due to the silent killer and asymptomatic nature of the diseases this may underestimate the risk factors of the disease. Though, we use the standardized WHO STEPs risk factor questionnaire allows for comparability on the presence of risk factors between various communities, regions, and countries.

## Conclusion

There was a high prevalence of hypertension among adults in the city of Addis Ababa, which may indicate a hidden epidemic in the population. Even though the study was conducted in the capital city, there was a large proportion of hypertensive respondents (61.95%) were unaware of having the condition and newly screened for the first time by the current study. Increasing age, gender being male, obesity and abdominal obesity, consumption of low vegetables, and raised triglyceride levels were significantly associated with hypertension.

As a result, lifestyle changes and the introduction of obesity and hypertension screening programs are recommended. These programs should focus on lifestyle changes, including eating fruits and vegetables, maintaining a normal weight, and weight loss intervention. The findings also underscore the vital need for community-based screening programs for the early detection of hypertension and obesity.

## Supporting information

**S1 File.**
(SAV)

**S1 Annex. English version questionnaires.**
(PDF)

**S2 Annex. Amharic version questionnaire.**
(PDF)

## Acknowledgments

The authors would like to express their appreciation to the following organizations and individuals for contributing to the success of this study: The authors wish to extend their most sincere thanks to all participants in the study. We thank the data collectors and supervisors. We also want to convey our deepest gratitude to Armed Forces Comprehensive Specialized Hospital because they allowed us to do all the lipid profiles with their laboratory technicians.

## Author Contributions

**Conceptualization:** Meseret Molla Asemu, Alemayehu Worku Yalew.

**Data curation:** Meseret Molla Asemu, Desalew Mekonnen.

**Formal analysis:** Meseret Molla Asemu.

**Funding acquisition:** Meseret Molla Asemu.

**Investigation:** Meseret Molla Asemu.

**Methodology:** Meseret Molla Asemu, Alemayehu Worku Yalew.

**Project administration:** Negussie Deyessa Kabeta.

**Resources:** Meseret Molla Asemu.

**Software:** Meseret Molla Asemu.

**Supervision:** Meseret Molla Asemu, Alemayehu Worku Yalew, Negussie Deyessa Kabeta.

**Validation:** Alemayehu Worku Yalew.

**Visualization:** Meseret Molla Asemu.

**Writing – original draft:** Meseret Molla Asemu.

**Writing – review & editing:** Meseret Molla Asemu, Alemayehu Worku Yalew, Negussie Deyessa Kabeta, Desalew Mekonnen.

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
