## [Decision Letter · Decision Letter 0]

21 Dec 2020

PONE-D-20-26679

Prevalence and risk factors of hypertension among adults: a Community Based Study in Addis Ababa, Ethiopia

PLOS ONE

Dear Dr. MESERET MOLLA ASEMU,

Thank you for submitting your manuscript to PLOS ONE. After careful consideration, we feel that it has merit but does not fully meet PLOS ONE’s publication criteria as it currently stands. Therefore, we invite you to submit a revised version of the manuscript that addresses the points raised during the review process.

As you will recognize from the comments of the reviewers both raised major points of critique, especially regarding design of the study and presentation of data.

Please submit your revised manuscript within 2 months. If you will need more time than this to complete your revisions, please reply to this message or contact the journal office at plosone@plos.org. Please include the following items when submitting your revised manuscript:

We look forward to receiving your revised manuscript.

Kind regards,

Rudolf Kirchmair

Academic Editor

PLOS ONE

Journal Requirements:

4. In the Methods, please discuss whether and how the questionnaire was validated and/or pre-tested. If this did not occur, please provide the rationale for not doing so.

7. Thank you for submitting the above manuscript to PLOS ONE. During our internal evaluation of the manuscript, we found significant text overlap between your submission and the following previously published works, some of which you are an author.

https://bmccardiovascdisord.biomedcentral.com/articles/10.1186/1471-2261-12-113

https://ejcm.journals.ekb.eg/article_11046_f56232e3d004cc38fe78b7b616f2799e.pdf

https://www.scribd.com/doc/115910728/Ncd-Report-Full-en-English

https://www.ncbi.nlm.nih.gov/pmc/articles/PMC2736927/

https://link.springer.com/article/10.1186/s12889-015-2610-8?code=241bf12b-10c4-493b-805c-c06d7a2cbf80

Please revise the manuscript to rephrase the duplicated text, cite your sources, and provide details as to how the current manuscript advances on previous work. Please note that further consideration is dependent on the submission of a manuscript that addresses these concerns about the overlap in text with published work.

Reviewers' comments:

Reviewer's Responses to Questions

**Comments to the Author**

1. Is the manuscript technically sound, and do the data support the conclusions?

Reviewer #1: Partly

Reviewer #2: Yes

2. Has the statistical analysis been performed appropriately and rigorously? 

Reviewer #1: No

Reviewer #2: Yes

3. Have the authors made all data underlying the findings in their manuscript fully available?

Reviewer #1: Yes

Reviewer #2: Yes

4. Is the manuscript presented in an intelligible fashion and written in standard English?

Reviewer #1: Yes

Reviewer #2: Yes

5. Review Comments to the Author

Reviewer #1: Line 71- Which are the "Non Communicable Diseases risk factors?

Line 164- Why p-value of < 0.20 was used as criteria to include it in the multivariable logistic

regression model?

Quite a small group of the study population were smokers in this study- can you explain why?

It is recommended that the diagnosis of hypertension should be based on:

repeated office BP measurements on more than one visit in the ESC-guidelines from 2018-

in this study the definition hypertension was defined on just one visit. Is the definition of hypertension chooses too weakly in this study?

Reviewer #2: Manuscript ID number:

PONE-D-20-26679

Title of paper:

Prevalence and risk factors of hypertension among adults: a Community Based Study in Addis Ababa, Ethiopia

Evaluation

Despite careful approach to investigate Prevalence and risk factors of hypertension, manuscript needs minor revisions to make it easy to understand before being published.

General comments:

1. Language editing strongly recommended

2. The body of the text suffers from several spelling and grammatical errors. Please consider a professional language edit. Example: scare (page 3 first paragraph),

3. Standardized your tables by removing the boarders and include P values in table 3

Page 2

4. In the abstract result section, almost all (96.2%) of participants consume vegetables and or fruits less than five times per day.

Is that feasible consuming vegetable & fruits five times per day in Ethiopian context? Or you mean five times per week? Make it clear

Page 3 & 4

5. Moreover, in Ethiopia non-communicable diseases such as hypertension and diabetes mellitus appear on the list of leading causes of morbidity and mortality in the hospitals and regional health bureaus across the country. A report by Ethiopian Public Health Institute (EPHI) in 2016 showed that 95% of the Ethiopian adult populations have 1-2 Non-Communicable Diseases risk factors (6, 7). But, there were scare data with regard to the magnitude and risk factors of common non communicable disease at the community level in Ethiopia including the study area Addis Ababa. Moreover, the study area represents the largest urban center in Ethiopia, hosting about 25% of the urban population in the country (5).

Since you are not intended to study all types of non-communicable diseases better to focus on hypertension). Paragraph 4, page 3 needs both language & grammatical edition.

Page 5

6. The method section, selection of the study participant,

the last paragraph a total of 3724 all needs to reconsider again

page 4

7. A community based cross-sectional study was conducted from June to October 2018 in Addis.

Please provide more precise date of study begin and termination

Page 5

8. Multi-stage cluster sampling techniques was employed. Seven of the ten sub-cities were selected purposefully by considering the area that was found, the population density and the economic activities.

You didn’t say anything about how you determine the sample size. How you calculate your sample size, what assumptions you used to calculate your sample size both for the magnitude & factors. Also, important you should show us how you allocate the number of participants to Sub-cities or Woreda Or Kebeles, Ketenas & households?

Page 5

9. One of the methods of maintaining the quality of data is keeping the data collection instrument valid & reliable (in you case weight scale & BP apparatus, the STEPS Questionnaire). In this regard you didn’t say anything.

How you maintain the reliability & validity of this instruments? We need more clarification on this issue

Page 9

10. In the description of the study participants, result section, you calculate both the mean with SD and Median with IQR for the respondents’ age.

What was the reason and which one was appropriate for your data? Need clarification

Page 11

11. In Tobacco use section to told us about 4.2% (150), of the survey participants were current smokers (daily and non-daily smokers) again in the last two sentence of the same section you presented, fifty-five (1.61%) were ever smoked cigarettes and One hundred nineteen (3.4%) were passive smoking or second-hand smoke.

What does this imply? Are these 55 peoples being among 150 who currently smoke? Needs to be clarified.

Page 13

12. Weight and height measurement were taken from all participants 3560 and the BMI was calculated for those participants. But you didn’t show how you calculate the BMI (only you defined BMI in the operational definition).

It is important to show how was the BMI calculated in the methods section. The procedure you used needs to be clearly kept in the method section

Page 13

13. You told us that blood sample was collected from 20% of the total study participants.

It is not sufficient to write 20% of total you need to write the actual number of participants you collect blood sample.

Page 13

14. In the result section, prevalence of hypertensin, you presented the overall prevalence of hypertension was 29.24% (95% CI: 27.75-30.74), slightly higher among men 30.13 (95% CI: 27.82-32.44), than women 28.58 (95% CI: 26.66-30.54) even though the difference was not statistically significant (χ2=1.015, P= 0.314).

But in the factors associated with you stated that sex had significant association with hypertension (The odds of hypertension was almost two times higher AOR= 1.88 (95% CI: 1.18-2.99) among males as compared to females). Needs clarification and reconsideration.

Page 19 discussion section

15. Hypertension is an important modifiable risk factor for cardiovascular disease (CVD). It currently accounts for about 13.5% of annual global deaths. Hypertension is directly responsible for 54% of all strokes and 47% of all coronary heart disease worldwide. Moreover, over half of this burden occurs in individuals aged 45–69 years, which is the most productive segment of the population (31).

Better to start your discussion by summary of your results and good if you use this in the introduction section

Page 19

16. …………. So, the prevalence of hypertension in the current study is slightly higher among men than women which is comparable with a community based study conducted in Addis Ababa, Ethiopia which reported the prevalence of hypertension was 31.5% and 28.9% among males and females, respectively (5). Moreover, this study is also comparable with other community-based studies conducted in Jalalabad, Afghanistan (28.4), Kenya (29.4%), Uganda 375 (30.5%), and Gondar city (28.3%) (12-15).

Here first you talk about the association between hypertension and gender or sex and on the next paragraph back to compare the prevalence with other studies. I see some confusion here I think you would want to change the order of the paragraph?

Page 20

17. ……… which the risk of hypertension increases with age. This is mainly due to systolic blood pressure increase with age, mainly because of reduced elasticity (increased stiffness) of the large conduit arteries (26). In this study respondents aged 30-49 years; had 3 times higher risk of hypertension and even moreover, it is 8 times higher risk among participants aged 50 years and above.

What is your message here for the patients and health care providers you provide? Is there anything that recommend to tackle this problem or age? You should better to emphasize on modifiable factors than non-modifiable like age & sex. Need your consideration

Page 20,

18. This finding (especially obese category) was in line with previous reports from Ethiopia, Kenya, Uganda, Sudan, Bangladesh, and Cameroon (13, 15, 17, 18, 21, 22, 24, 25, 27). Moreover, the risk of hypertension was 2 times higher among abdominally obese respondents and this finding is in line with other studies (24, 25, 28, 29) and the same to the level of triglyceride also.

Since this is the most important area that your recommendation is focused, comparing the findings is not sufficient. Better to find the reason of similarity or differences and give your recommendation or message based on that. Therefore, you need to work on it and put your recommendation.

Page 21, first paragraph

19. In contradiction, in this study the above variables were not significantly associated with hypertension. The inconsistency of these findings may be due to the low prevalence of these factors in the community especially among females.

What does it mean? I don’t think your reason for differences is correct. May you need to find tangible reason for this difference.

Page 21

20. Additionally, the respondents might not know whether they had a family history of hypertension or diabetes due to the silent killer nature of the diseases this may underestimate the prevalence of the diseases.

How the silent killer nature of the disease affects the prevalence of hypertension since the prevalence was determined by measuring their blood pressure? Or you want to say the severity of the disease? Not clear

Do you think diabetes is a silent killer? Since your objectives did not include diabetes why you include here?

Page 21

21. The other reason should since some of the information was based on self-report and is subjected to social desirability and recall biases.

These issues are very critical in research. How you manage this social desirability and recall biases since this can affect severely your findings? You have to show us either in the discussion or method section how you control theses biases clearly? In addition, with all these short comings or limitations do think your research could be eligible for publication? Better to avoid those limitations that can be controlled methodologically

Page 21

22. In the conclusion section …. There was a high prevalence of hypertension among adults in the Addis Ababa city and this may show a hidden epidemic in the population. What is your reference to say high prevalence or to conclude this is a hidden epidemic? You have to show here

6. PLOS authors have the option to publish the peer review history of their article (what does this mean?). If published, this will include your full peer review and any attached files.

Reviewer #1: No

Reviewer #2: No

---

## [Author Response · Author response to Decision Letter 0]

6 Feb 2021

A rebuttal letter

Manuscript PONE-D-20-26679

Response to Reviewers

Dear Rudolf Kirchmair, 

Thank you for the opportunity to provide a revised version of the manuscript. “Prevalence and risk factors of hypertension among adults: a Community Based Study in Addis Ababa, Ethiopia” for publication in PLOS ONE Journal. We appreciate the time and effort you and the examiners put into providing comments on our manuscript. We have incorporated the suggestions and comments made by the reviewers. These changes are highlighted in the manuscript. A point-by-point response to the reviewers’ comments and concerns is provided below in blue.

PONE-D-20-26679

Prevalence and risk factors of hypertension among adults: a Community Based Study in Addis Ababa, Ethiopia

PLOS ONE

Dear Dr. MESERET MOLLA ASEMU,

Thank you for submitting your manuscript to PLOS ONE. After careful consideration, we feel that it has merit but does not fully meet PLOS ONE’s publication criteria as it currently stands. Therefore, we invite you to submit a revised version of the manuscript that addresses the points raised during the review process.

As you will recognize from the comments of the reviewers both raised major points of critique, especially regarding design of the study and presentation of data.

Please submit your revised manuscript within 2 months. If you will need more time than this to complete your revisions, please reply to this message or contact the journal office at plosone@plos.org. Please include the following items when submitting your revised manuscript:

We look forward to receiving your revised manuscript.

Kind regards,

Rudolf Kirchmair

Academic Editor

PLOS ONE

Journal Requirements:

 Authors’ response: Dear academic editor, thank you for providing the link. We carefully read and edited our manuscript as per the guidelines.

 Authors’ response: Thank you for your feedback and suggestion. We accepted the comments on language usage, spelling, and grammar, based on the comments we edited our manuscript as much as possible by using online grammar and language checkers (Grammarly) and with my friend speaks fluent English at our university. We have prepared and attached our manuscript highlighting the changes and uploaded it as a *supporting information* file. We have also prepared and attached the edited manuscript and uploaded it as the new *manuscript* file.

 Authors’ response: Thank you for your comment. We have attached the questionnaires we used for the current survey in both the original (Amharic) and English language as Supporting Information.

4. In the Methods, please discuss whether and how the questionnaire was validated and/or pre-tested. If this did not occur, please provide the rationale for not doing so.

 Authors’ response: Thank you for your feedback. This was incorporated into the method portion of our manuscript. The questionnaire was also adapted from the World Health Organization and validated in various previous studies in Ethiopia. However, pretesting took place.

 Authors’ response: Thank you for your comment. The current study is part of a large study with multiple objectives to assess Epidemiology of common non communicable diseases, among adults in Addis Ababa, Ethiopia. Further publication is expected from the dataset which prevents us from making it publicly right now. So, we made changes in our cover letter and we have included in the updated Data Availability statement part.

 Authors’ response: Thank you for your comment. We ensured that we have an ORCID iD.

7. Thank you for submitting the above manuscript to PLOS ONE. During our internal evaluation of the manuscript, we found significant text overlap between your submission and the following previously published works, some of which you are an author.

https://bmccardiovascdisord.biomedcentral.com/articles/10.1186/1471-2261-12-113

https://ejcm.journals.ekb.eg/article_11046_f56232e3d004cc38fe78b7b616f2799e.pdf

https://www.scribd.com/doc/115910728/Ncd-Report-Full-en-English

https://www.ncbi.nlm.nih.gov/pmc/articles/PMC2736927/

https://link.springer.com/article/10.1186/s12889-015-2610-8?code=241bf12b-10c4-493b-805c-c06d7a2cbf80

Please revise the manuscript to rephrase the duplicated text, cite your sources, and provide details as to how the current manuscript advances on previous work. Please note that further consideration is dependent on the submission of a manuscript that addresses these concerns about the overlap in text with published work.

 Authors’ response: Thank you for your comment. We have redrafted the entire duplicate text into the manuscript based on your recommendation.

[Note: HTML mark up is below. Please do not edit.]

Reviewers' comments:

Reviewer's Responses to Questions

Comments to the Author

1. Is the manuscript technically sound, and do the data support the conclusions?

Reviewer #1: Partly

Reviewer #2: Yes

 Authors’ response: Thank you for your comment. The manuscript is revised accordingly to improve its scientific writing. The conclusions are revised as well to reflect the data presented. 

2. Has the statistical analysis been performed appropriately and rigorously? 

Reviewer #1: No

Reviewer #2: Yes

 Authors’ response: Thank you for your comment. We accepted your comment, and we have incorporated it into the result part of our manuscript. We have included the p value in table 3, as suggested by the reviewer. Moreover, we also incorporated other comments from the reviewers. 

3. Have the authors made all data underlying the findings in their manuscript fully available?

Reviewer #1: Yes

Reviewer #2: Yes

 Authors’ response: Thank you for your comments. The current study is part of a large study with multiple objectives to assess Epidemiology of common non communicable diseases, among adults in Addis Ababa, Ethiopia. Although additional publications are planned based on the dataset, we have included the raw data in SPSS format in the data availability section.

4. Is the manuscript presented in an intelligible fashion and written in standard English?

Reviewer #1: Yes

Reviewer #2: Yes

 Authors’ response: Thank you for your comments. The manuscript has been reassessed and all grammatical mistakes have been corrected. 

5. Review Comments to the Author

Reviewer #1: Line 71- Which are the "Non Communicable Diseases risk factors?

Authors’ response: Thank you for your question. The 1-2 non communicable risk factors were over-weight (BMI ≥ 25 kg/m2), consumption of fruit and vegetables less than 5 servings per day and raised BP (SBP≥140 and/or DBP ≥ 90 mmHg or currently on medication for raise blood pressure and insufficient physical activities.

Line 164- Why p-value of < 0.20 was used as criteria to include it in the multivariable logistic

regression model?

Authors’ response: Thank you for your question. We read different articles and as a rule of thumb, they selected all the variables whose p-value < 0.2 on binary logistic regression for multivariable logistic regression. But we could not obtain from the standard biostatistics books, so corrected by entering all the variables we used in binary logistics analysis into multivariable logistic regression.

Quite a small group of the study population were smokers in this study- can you explain why?

Authors’ response: Thank you for your question. As we have seen from different studies conducted in Ethiopia, including the study area, Addis Ababa, the prevalence of smoking was low; the possible reason may be the number of smokers in the study setting was low.

It is recommended that the diagnosis of hypertension should be based on:

repeated office BP measurements on more than one visit in the ESC-guidelines from 2018-

in this study the definition hypertension was defined on just one visit. Is the definition of hypertension chooses too weakly in this study?

Authors’ response: Thank you for your comment. We have measured the blood pressure of study participants three times, and we took the mean of the second and the third records because mostly the first record became high. The World Health Organization; and the American Health Association recommends one visit three times measurements to define hypertension during a community survey. The definition is not weak because we measured three times; moreover, we measured their blood pressure in their home; this also minimizes the white coat false records of high blood pressure. Also, the literature that we used in the discussion part used this method.

Reviewer #2: Manuscript ID number:

PONE-D-20-26679

Title of paper:

Prevalence and risk factors of hypertension among adults: a Community Based Study in Addis Ababa, Ethiopia

Evaluation

Despite careful approach to investigate Prevalence and risk factors of hypertension, manuscript needs minor revisions to make it easy to understand before being published.

General comments:

1. Language editing strongly recommended

Authors’ response: Thank you for your feedback and suggestion. We accepted the comments and strongly recommended issues on language editing and based on the comments we thoroughly edited our manuscript as much as possible by using online grammar and language checkers (Grammarly) and with my friend speaks fluent English in our university.

2. The body of the text suffers from several spelling and grammatical errors. Please consider a professional language edit. Example: scare (page 3 first paragraph),

Authors’ response: Thank you for your feedback and suggestion. We accepted the comment and made corrections.

3. Standardized your tables by removing the boarders and include P values in table 3

Page 2

Authors’ response: Thank you for your feedback and suggestion. The comment was accepted and the p-values were added to Table 3. 

4. In the abstract result section, almost all (96.2%) of participants consume vegetables and or fruits less than five times per day.

Is that feasible consuming vegetable & fruits five times per day in Ethiopian context? Or you mean five times per week? Make it clear

Page 3 & 4

Authors’ response: Thank you for your clarification question. Healthy eating, including an adequate intake of fruits and vegetables (five servings a day), is one of the key public health measures to prevent NCDs. Eating fruits and vegetables five times daily is recommended by the World Health Organization in developed and developing countries, including Ethiopia. But, in many countries worldwide, the vast majority of the population consumes less than the recommended amount of five servings of fruit and vegetables per day, though low intake of fruits and vegetables was estimated to cause 4.7% of the global disease burden – as estimated in DALYs. And in our study, as we mentioned in the method part we used the WHO STEPS instrument and one of the major core questions was to assess the dietary habit, including the fruits and vegetables of participants whether they are in line with the WHO recommendation or not. 

5. Moreover, in Ethiopia non-communicable diseases such as hypertension and diabetes mellitus appear on the list of leading causes of morbidity and mortality in the hospitals and regional health bureaus across the country. A report by Ethiopian Public Health Institute (EPHI) in 2016 showed that 95% of the Ethiopian adult populations have 1-2 Non-Communicable Diseases risk factors (6, 7). But, there were scare data with regard to the magnitude and risk factors of common non communicable disease at the community level in Ethiopia including the study area Addis Ababa. Moreover, the study area represents the largest urban center in Ethiopia, hosting about 25% of the urban population in the country (5).

Since you are not intended to study all types of non-communicable diseases better to focus on hypertension). Paragraph 4, page 3 needs both language & grammatical edition.

Page 5

Authors’ response: Thank you for your comments. We accepted the comment and made corrections. 

6. The method section, selection of the study participant,

the last paragraph a total of 3724 all needs to reconsider again

page 4

Authors’ response: Thank you for your comment. We accepted the comment and made corrections. 

7. A community based cross-sectional study was conducted from June to October 2018 in Addis.

Please provide more precise date of study begin and termination

Page 5

Authors’ response: Thank you for your comment. We accepted the comment and incorporated and re-wrote the exact start and end date of the study.

8. Multi-stage cluster sampling techniques was employed. Seven of the ten sub-cities were selected purposefully by considering the area that was found, the population density and the economic activities.

You didn’t say anything about how you determine the sample size. How you calculate your sample size, what assumptions you used to calculate your sample size both for the magnitude & factors. Also, important you should show us how you allocate the number of participants to Sub-cities or Woreda Or Kebeles, Ketenas & households?

Page 5

Authors’ response: Thank you for your comment. We accept the comment, and we have incorporated the sample size determination. We also explained how we allocate the number of participants in the selected sub-cities in the method part of our manuscript.

9. One of the methods of maintaining the quality of data is keeping the data collection instrument valid & reliable (in you case weight scale & BP apparatus, the STEPS Questionnaire). In this regard you didn’t say anything.

How you maintain the reliability & validity of this instruments? We need more clarification on this issue

Page 9

Authors’ response: Thank you for your questions. We agree with the question and have responded to it in the method portion of our manuscript.

10. In the description of the study participants, result section, you calculate both the mean with SD and Median with IQR for the respondents’ age.

What was the reason and which one was appropriate for your data? Need clarification

Page 11

Authors’ response: Thank you for your comment. We accepted the comment. Because our variable age was skewed, we chose the median as a measure of central tendency rather than as a mean. We have corrected in the result part of our manuscript.

11. In Tobacco use section to told us about 4.2% (150), of the survey participants were current smokers (daily and non-daily smokers) again in the last two sentence of the same section you presented, fifty-five (1.61%) were ever smoked cigarettes and One hundred nineteen (3.4%) were passive smoking or second-hand smoke.

What does this imply? Are these 55 peoples being among 150 who currently smoke? Needs to be clarified.

Page 13

Authors’ response: Thank you for your clarification question. From the total participants, 150 (4.2%) of them was currently a smoker. But if they were not current smokers, we asked them whether they smoke cigarettes or not by saying, “In the past, did you ever smoke any tobacco products?” If they said yes to the above question, we considered them as previous smokers or Ex-smoker. So, from the total current non-smokers, we got 55 participants, classified under the previous smoker; this number is not included in the 150 current smokers. 

12. Weight and height measurement were taken from all participants 3560 and the BMI was calculated for those participants. But you didn’t show how you calculate the BMI (only you defined BMI in the operational definition).

It is important to show how was the BMI calculated in the methods section. The procedure you used needs to be clearly kept in the method section

Page 13

Authors’ response: Thank you for your comment. We accepted your comment, and we have incorporated it into the method part of our manuscript.

13. You told us that blood sample was collected from 20% of the total study participants.

It is not sufficient to write 20% of total you need to write the actual number of participants you collect blood sample.

Page 13

Authors’ response: Thank you for your comment. We accepted your comment, and we have incorporated it into the result part of our manuscript.

14. In the result section, prevalence of hypertensin, you presented the overall prevalence of hypertension was 29.24% (95% CI: 27.75-30.74), slightly higher among men 30.13 (95% CI: 27.82-32.44), than women 28.58 (95% CI: 26.66-30.54) even though the difference was not statistically significant (χ2=1.015, P= 0.314).

But in the factors associated with you stated that sex had significant association with hypertension (The odds of hypertension was almost two times higher AOR= 1.88 (95% CI: 1.18-2.99) among males as compared to females). Needs clarification and reconsideration.

Page 19 discussion section

Authors’ response: Thank you for your comment. We accepted your comment, and we have corrected it into the result part of our manuscript. 

15. Hypertension is an important modifiable risk factor for cardiovascular disease (CVD). It currently accounts for about 13.5% of annual global deaths. Hypertension is directly responsible for 54% of all strokes and 47% of all coronary heart disease worldwide. Moreover, over half of this burden occurs in individuals aged 45–69 years, which is the most productive segment of the population (31).

Better to start your discussion by summary of your results and good if you use this in the introduction section

Page 19

Authors’ response: Thank you for your comment. We accepted your comment, and we have corrected it.

16. …………. So, the prevalence of hypertension in the current study is slightly higher among men than women which is comparable with a community based study conducted in Addis Ababa, Ethiopia which reported the prevalence of hypertension was 31.5% and 28.9% among males and females, respectively (5). Moreover, this study is also comparable with other community-based studies conducted in Jalalabad, Afghanistan (28.4), Kenya (29.4%), Uganda 375 (30.5%), and Gondar city (28.3%) (12-15).

Here first you talk about the association between hypertension and gender or sex and on the next paragraph back to compare the prevalence with other studies. I see some confusion here I think you would want to change the order of the paragraph?

Page 20

Authors’ response: Thank you for your comment. Your feedback is accepted and corrected.

17. ……… which the risk of hypertension increases with age. This is mainly due to systolic blood pressure increase with age, mainly because of reduced elasticity (increased stiffness) of the large conduit arteries (26). In this study respondents aged 30-49 years; had 3 times higher risk of hypertension and even moreover, it is 8 times higher risk among participants aged 50 years and above.

What is your message here for the patients and health care providers you provide? Is there anything that recommend to tackle this problem or age? You should better to emphasize on modifiable factors than non-modifiable like age & sex. Need your consideration

Page 20,

Authors’ response: Thank you for your comment. Our message for the patients was to be screened and get treatment. So, they can prevent complications associated with untreated hypertension. For the health providers, especially the Health Extension Workers (in our country, they went to each community house to deliver contraception, vaccine), we told them to take blood pressure whether they have a symptom or not for those aged peoples. The other thing we give more emphasis on modifiable factors since they can modify them.

18. This finding (especially obese category) was in line with previous reports from Ethiopia, Kenya, Uganda, Sudan, Bangladesh, and Cameroon (13, 15, 17, 18, 21, 22, 24, 25, 27). Moreover, the risk of hypertension was 2 times higher among abdominally obese respondents and this finding is in line with other studies (24, 25, 28, 29) and the same to the level of triglyceride also.

Since this is the most important area that your recommendation is focused, comparing the findings is not sufficient. Better to find the reason of similarity or differences and give your recommendation or message based on that. Therefore, you need to work on it and put your recommendation.

Page 21, first paragraph

Authors’ response: Thank you for your comment. Your feedback is accepted and corrected.

19. In contradiction, in this study the above variables were not significantly associated with hypertension. The inconsistency of these findings may be due to the low prevalence of these factors in the community especially among females.

What does it mean? I don’t think your reason for differences is correct. May you need to find tangible reason for this difference.

Page 21

Authors’ response: Thank you for your comment. Your feedback is accepted and corrected. 

20. Additionally, the respondents might not know whether they had a family history of hypertension or diabetes due to the silent killer nature of the diseases this may underestimate the prevalence of the diseases.

How the silent killer nature of the disease affects the prevalence of hypertension since the prevalence was determined by measuring their blood pressure? Or you want to say the severity of the disease? Not clear

Do you think diabetes is a silent killer? Since your objectives did not include diabetes why you include here?

Page 21

Authors’ response: Thank you for your comment. Your feedback is accepted and corrected. Additionally, the respondents might not know whether they had a family history of hypertension due to the silent killer nature of the diseases this may underestimate the risk factors of the diseases. Since family history of hypertension is one of the risk factors of hypertension.

21. The other reason should since some of the information was based on self-report and is subjected to social desirability and recall biases.

These issues are very critical in research. How you manage this social desirability and recall biases since this can affect severely your findings? You have to show us either in the discussion or method section how you control theses biases clearly? In addition, with all these short comings or limitations do think your research could be eligible for publication? Better to avoid those limitations that can be controlled methodologically

Page 21

Authors’ response: Thank you for your comment. Your feedback is accepted and corrected.

We used different mechanisms to avoid biases. To avoid social desirability bias, first, we explained the aim of the survey for each participant and during data collection; we kept it anonymous and confidential. After data collection, the information is kept in a safe and secured place. Moreover, to avoid recall bias we asked timeline timeliness of the information and standard questionnaires prepared by the World Health Organization. So, since we did all the activities that help us to minimize the biases we excluded the sentence included as a limitation

22. In the conclusion section …. There was a high prevalence of hypertension among adults in the Addis Ababa city and this may show a hidden epidemic in the population. What is your reference to say high prevalence or to conclude this is a hidden epidemic? You have to show here

Authors’ response: Thank you for your question. Our finding showed that 30% of study participants had hypertension but a study conducted in one of the urban areas of Ethiopia showed that the prevalence of hypertension was 20%; moreover, a large proportion, 62% of them unaware of having the problem; that is why we would like to say this showed the hidden epidemic of the disease among adults aged 18 year and above. Moreover, hypertension is a chronic disease if it is not diagnosed and treated early may end up with life-threatening complication and death.

6. PLOS authors have the option to publish the peer review history of their article (what does this mean?). If published, this will include your full peer review and any attached files.

Do you want your identity to be public for this peer review? For information about this choice, including consent withdrawal, please see our Privacy Policy.

Reviewer #1: No

Reviewer #2: No

 Authors’ response:Thank you. We agreed and corrected that comment.

Authors’ response:Thank you. We have registered with PACE and all the tables do fit with PLOS. We downloaded from the PACE and uploaded as Table 1, Table 2 and Table 3 in TIF format.

---

## [Decision Letter · Decision Letter 1]

9 Mar 2021

Prevalence and risk factors of hypertension among adults: a Community Based Study in Addis Ababa, Ethiopia

PONE-D-20-26679R1

Dear Dr. ASEMU,

We’re pleased to inform you that your manuscript has been judged scientifically suitable for publication and will be formally accepted for publication once it meets all outstanding technical requirements.

Kind regards,

Rudolf Kirchmair

Academic Editor

PLOS ONE

Additional Editor Comments (optional):

Reviewers' comments:

Reviewer's Responses to Questions

**Comments to the Author**

1. If the authors have adequately addressed your comments raised in a previous round of review and you feel that this manuscript is now acceptable for publication, you may indicate that here to bypass the “Comments to the Author” section, enter your conflict of interest statement in the “Confidential to Editor” section, and submit your "Accept" recommendation.

Reviewer #2: All comments have been addressed

2. Is the manuscript technically sound, and do the data support the conclusions?

Reviewer #2: Yes

3. Has the statistical analysis been performed appropriately and rigorously? 

Reviewer #2: Yes

4. Have the authors made all data underlying the findings in their manuscript fully available?

Reviewer #2: Yes

5. Is the manuscript presented in an intelligible fashion and written in standard English?

Reviewer #2: Yes

6. Review Comments to the Author

Reviewer #2: The Author tried to address More or less the comments given by me. It can be published on your journal

7. PLOS authors have the option to publish the peer review history of their article (what does this mean?). If published, this will include your full peer review and any attached files.

Reviewer #2: **Yes: **Daniel G/Tsadik W/giorgis

---

## [Editor Report · Acceptance letter]

11 Mar 2021

PONE-D-20-26679R1 

Prevalence and risk factors of hypertension among adults: a Community Based Study in Addis Ababa, Ethiopia 

Dear Dr. Asemu:

I'm pleased to inform you that your manuscript has been deemed suitable for publication in PLOS ONE. Congratulations! Your manuscript is now with our production department. 

Kind regards, 

on behalf of

Prof Rudolf Kirchmair 

Academic Editor

PLOS ONE